# Concomitant Guillain–Barré Syndrome and COVID-19: A Meta-Analysis of Cases

**DOI:** 10.3390/medicina58121835

**Published:** 2022-12-13

**Authors:** Skylar A. Bentley, Sarfraz Ahmad, Firas H. Kobeissy, Hale Z. Toklu

**Affiliations:** 1College of Health Sciences, University of Central Florida, Orlando, FL 32816, USA; 2Burnette Honors College, University of Central Florida, Orlando, FL 32816, USA; 3College of Medicine, University of Central Florida, Orlando, FL 32827, USA; 4Advent Health Cancer Institute, Orlando, FL 32804, USA; 5Department of Emergency Medicine, College of Medicine, University of Florida, Gainesville, FL 32608, USA

**Keywords:** SARS-CoV-2, COVID-19, Guillain–Barré syndrome, GBS, meta-analysis, neurological complications

## Abstract

*Background and Objectives*: Recent findings demonstrate that the transmigration of severe acute respiratory syndrome Coronavirus 2 (SARS-CoV-2) to the nervous system implicates severe neurotropic pathologies, including the onset of the rare disease called Guillain–Barré syndrome (GBS) which is characterized by immune-mediated polyneuropathy. This study aimed to identify the predisposing factors and the clinical features of coronavirus disease 2019 (COVID-19)-induced GBS. *Materials and Methods*: We have performed an analysis of 147 cases. A systematic review of the published research work was performed per the PRISMA statement to obtain individual participant data (IPD) for the meta-analysis. The search was conducted through PubMed, using the combined search terms “Guillain–Barré syndrome” and “COVID-19”. All case reports and series in the English language with accessed full text were included in the search. *Results*: A systematic database search led to the retrieval of 112 peer-reviewed articles published between 1 April 2020, and 8 February 2022. The articles comprised 16 case series and 96 case reports containing IPD for 147 patients. Our findings showed that 77.6% of all cases were 40 years or older. Males comprised most of the cases (65.3%; *n* = 96). The intensive care unit (ICU) admission was 44.9%, and the need for mechanical ventilation (MV) was 38.1%. The patients presented with hyporeflexia or areflexia (84.4%; *n* = 124), lower limb strength and sensation impairment (93.2%; *n* = 138), upper limb strength and sensation impairment (85.7; *n* = 126), and somatic sensation impairment (72.8%; *n* = 107). The patients presented with increased cerebral spinal fluid (CSF) protein levels (92%; *n* = 92) and the presence of CSF albuminocytological dissociation (83.5%; *n* = 71). The most common variant of GBS observed was acute inflammatory demyelinating polyneuropathy (AIDP). We found that predisposing factors concomitant with COVID-19 and GBS were male gender and older age. Among the cases, patient mortality was 10.9%. *Conclusions*: A gap of knowledge exists regarding the complete spectrum of clinical characteristics of COVID-19-related GBS. Recent findings suggest that SARS-CoV-2 triggers GBS, as it follows a similar para-infectious pattern as the other viral agents contributing to the onset of GBS.

## 1. Introduction

The current disease outbreak caused by severe acute respiratory syndrome Coronavirus 2 (SARS-CoV-2) has become a global health emergency. Although respiratory impairment is the main symptom associated with the pathology of the Coronavirus disease 2019 (COVID-19), there are reports of neurological manifestations associated with the disease. Opportunistic viral pathogens, such as human Coronaviruses, may spread into other tissues, such as the central nervous system (CNS), where additional pathologies may be induced [1]. The family of β-Coronaviruses to which SARS-CoV-2 belongs has previously been identified within the brain (especially the brainstem) [2]. The mechanism of neuroinvasion by SARS-CoV-2 is still poorly understood, and it is imperative to understand this phenomenon, as the neurological manifestations of COVID-19 are of growing concern. 

Several pathways for the invasion of SARS-CoV-2 to the CNS have been postulated. Transmigration of SARS-CoV-2 to the brain may occur via the olfactory pathway, general circulation, or the peripheral neurons of the lungs [2]. The transmigration of SARS-CoV-2 to the nervous system implicates severe neurological pathologies such as ischemic changes of neurons, demyelination of nerve fibers, and diseases such as polyneuropathy, encephalitis, and aortic ischemic stroke [3]. In recent months, there have been increasing reports which show the association and para-infectious nature between Guillain–Barré syndrome (GBS) and SARS-CoV-2 [4]. It is postulated that COVID-19 triggers the onset of GBS similarly to cytomegalovirus, the Epstein–Barr virus, the Middle East Respiratory Syndrome (MERS), Hepatitis E, and the Zika virus contribute to the etiology of GBS through autoimmune dysregulation and augmentation of the cytokine release storm (CRS) [5]. Many of the cytokines that participate in the pathogenesis of COVID-19 are also involved in the onset of GBS and play a pivotal role(s) in the rapid progression of GBS [6]. 

Details such as the type and severity of the preceding infection and patient-related host factors are the prime determinants of the onset, phenotypic form, and progression of GBS [7]. Multidisciplinary care for patients with GBS is imperative to manage the potentially severe and life-threatening complications associated with its onset and progression [7]. Plasma exchange (a well-supported treatment for GBS) removes neurotoxic antibodies and other inflammatory mediators [7]. It was found that plasma exchange treatment yields optimal outcomes when performed within two to four weeks following the initial signs of weakness [7]. 

The relationship between SARS-CoV-2 and GBS is yet to be understood. Therefore, this study aimed to provide insight into the predisposing factors through sociodemographic characteristics of the patients and their clinical outcomes associated with concomitant GBS and COVID-19. 

## 2. Materials and Methods

### 2.1. Data Extraction

A systematic review of the published research work was performed per the Preferred Reporting Items for Systematic Reviews and Meta-Analyses (PRISMA) statement to obtain individual participant data (IPD) for the meta-analysis [8]. The following review protocols were registered through Prospero, an international prospective register of systematic reviews. The registration code CRD42022379581 can be used to find the review on Prospero. A literature search was carried out across several major databases such as PubMed, Cochrane, Medline, BioMed Central, Embase, Scopus, and Web of Science. The search supported that PubMed would be the most fitting database because it had the most significant number of articles closely related to our novel topic. It was confirmed that the other databases were less inclusive and yielded similar results to the PubMed database. A final extensive literature search was conducted through PubMed, using the combined search terms “Guillain–Barré syndrome” and “COVID-19”. Search years were limited to 2020, 2021, and 2022 and all case reports and series were included in the search. The articles with accessed full texts written in the English language were included. The meta-analyses, randomized controlled trials, and systemic reviews were excluded. Overlapping publications were reviewed and excluded. The cases that were unrelated to either GBS or COVID-19 were excluded. If the patients had a prior history of GBS prior to COVID-19, they were also excluded. Patients were excluded if an infectious agent other than SARS-CoV-2 was present in the blood or CSF. Patients were also required to have a confirmed COVID-19 diagnosis by either a reverse transcription–polymerase chain reaction (RT–PCR) test or the presence of serum antibodies. 

### 2.2. Data Elements and Outcomes

Normal reference values for laboratory data were established to maintain consistency when reporting data findings. Reference values for the laboratory data are displayed in Table A1 in Appendix A. The data was recorded through Microsoft Excel and coded to be transferred into IBM SPSS for statistical analysis. Table A2 in Appendix A displays the coding mechanism used for the project. The data elements included demographic information, comorbidities, laboratory tests, treatments, and procedures, including invasive mechanical ventilation, conducted during hospitalization. The clinical outcomes were evaluated by disease severity (intensive care unit (ICU) level of care, reliance on mechanical ventilation (MV), and vasopressor support), time on MV, hospital length-of-stay (LOS), and mortality. Anti-ganglioside antibodies, neurological findings, and GBS subtype were also recorded. The variants of GBS in the study included acute inflammatory demyelinating polyneuropathy (AIDP), acute motor and sensory axonal neuropathy (AMSAN), acute motor axonal neuropathy (AMAN), Miller Fisher syndrome (MFS), pharyngeal-cervical-brachial (PCB) variant, and facial diplegia (FD), hemiplegia, or weakness with paresthesia.

### 2.3. Ethics

The study does not involve research with human subjects and therefore is exempt from an institutional review board (IRB) oversight. Data were obtained from previously published studies in which the primary investigators obtained informed consent.

### 2.4. Statistical Analyses

The statistical analyses were performed using IBM SPSS Statistics Software for Windows version 28.0 (IBM Corp, Armonk, NY, USA). The data points included frequencies and descriptive statistics such as mean, standard deviation (SD), and Range. A Chi-square test was used to analyze the association in cross-tabulations. *p* < 0.05 value was considered statistically significant.

## 3. Results

### 3.1. Patient Population

The PRISMA flow diagram illustrating the exclusion of articles is presented in Figure A1 in Appendix B. These articles found through the literature search comprised 16 case series and 96 case reports containing IPD for 147 patients. Table A3 in Appendix C provides the author(s) name, year of publication, and the number of patients in this project’s included articles. 

The results for sociodemographic patient data and clinical outcome measures are outlined in Table 1. The patients were about two-thirds male (65.3%; *n* = 96), and about one-third were female (34.7%; *n* = 51). The mean age among the patients was 52 years old (*SD* = 18). There were 33 (22.4%) patients younger than 40, while most patients (77.6%; *n* = 114) were 40 years or older.

The patient’s hospital LOS ranged from one to 62 days and had a mean value of 23 days (*SD* = 17). There was nearly an equal ratio of patients that stayed in the hospital for less than 20 days (54.5%; *n* = 30) and patients that stayed in the hospital for 20 or more days (45.5%; *n* = 25). A similar relationship occurred between whether the patients required admission to the ICU. A slight majority of patients did not require ICU admission (55.1%; *n* = 81) compared to those admitted to the ICU (44.9%; *n* = 66). Roughly two-thirds of the patients did not require MV (61.9%; *n* = 91), while the remaining patients required MV due to respiratory failure (38.1%; *n* = 56). For patients who did require MV, the days they relied on MV before being extubated or expiring ranged from two to 26 days, with an average of 12 days (*SD* = 7). There was a high survival rate among patients, with 78.9% (*n* = 116) of patients that survived and 10.9% (*n* = 16) of patients that did not survive. Of the patients who survived, 10.2% (*n* = 15) were not yet discharged from the hospital to their homes or a rehabilitation center.

### 3.2. Serological Analyses

Serological studies examined the relationship between the onset of GBS and COVID-19 by identifying serum anti-ganglioside antibodies. Results are summarized in Figure 1. Only 18.8% (*n* = 9) of patients tested positive for serum anti-ganglioside antibodies, while most tested patients (81.3%; *n* = 39) were negative.

### 3.3. Cerbrospinal Fluid Analyses

The CSF analysis revealed that COVID-19 was detected in four cases of the 55 tested (6.8%). Protein levels in CSF were elevated in 92% of the patients (*n* = 92), as displayed in Figure 2. Figure 2 also displays that only 7% (*n* = 7) of the patients had a standard CSF protein value, while only one percent had a low CSF protein value. The most common CSF protein value range was between 51-75 mg/dL, comprising 30% (*n* = 30) of the values among 100 total patients. The white blood cell (WBC) count was within the normal limits in 94.2% (*n* = 81) of the total cases (*n* = 86). The total WBC count obtained from 55 patients ranged from zero to 18 cells/µL of blood and had a mean value of 2.3 cells/µL (*SD* = 3.1). Cerebrospinal fluid albuminocytological dissociation (elevated CSF total protein value without pleocytosis) was observed in 83.5% (*n* = 71) of cases. The CSF glucose values were abnormal in 52.9% (*n* = 18) of the 34 cases. The CSF glucose values recorded among 34 patients ranged from 50 to 166 mg/dL and had a mean value of 76.4 mg/dL (*SD* = 22.5).

### 3.4. Neurophysiological Findings

The neurophysiological symptoms reported by each study are included in Table 2. The neurological manifestations reported from a total of 147 cases are as follows: 7.5% report an abnormal plantar response, 12.9% report aphasia, 46.3% report ataxia, 20.4% report dysphagia, 42.2% report facial palsy, weakness, or plegia, 4.1% fecal incontinence, 10.9% report urinary difficulties 17.7% report hypogeusia or ageusia, 84.4% report hyporeflexia or areflexia, 15.6% report hyposmia or anosmia, 72.8% report impaired somatic sensation, 15% report lumbar pain, 23.8% report myalgia, and 7.5% report neck flexion weakness. 

### 3.5. Limb Strength and Sensation

Patients presented with both upper and lower limb strength and sensation abnormalities. The findings are displayed in Figure 3. All 147 cases included data on the upper and lower limb abnormalities. A vast majority of cases reported that the patient’s lower limbs were affected by either weakness, plegia, or paresthesia (93.2%; *n* = 137). Out of the 126 cases that reported upper limb abnormalities, 44 reported both paresthesia and weakness, 20 reported complete upper limb plegia without paresthesia, 50 reported weakness without paresthesia, and 11 reported paresthesia without weakness. In all cases involving limb plegia, it was assumed that patients with limb plegia also presented with limb weakness. Out of the 137 cases that reported lower limb abnormalities, 60 reported both paresthesia and weakness, 24 reported complete lower limb plegia without paresthesia, 48 reported weakness without paresthesia, and five reported paresthesia without weakness. As for the results regarding upper limb abnormalities, most cases (85.7%; *n* = 126) reported that the patient’s upper limbs were affected by either weakness, plegia, or paresthesia.

### 3.6. Electrophysiological Findings

Electrophysiological studies were conducted in many studies to explore the distal latency (ms), conduction velocity (m/s), amplitudes (mV for motor and μV for sensory), onset latency, peak latency, and F-response latency of the sensory and motor nerves. Specific nerves commonly evaluated between studies included the ulnar, peroneal, tibial, and sural nerves. Sensory nerve action potential (SNAP) and compound muscle action potential (CMAP) test results are displayed in Figure 4. The SNAP tests were shown to have abnormal values in 86.9% (*n* = 86) of cases. The CMAP tests also revealed that most patients received abnormal test results (99%; *n* = 102). 

### 3.7. Guillain–Barré Syndrome Subtype Classification

A total of 85 cases included the GBS subtype associated with the patient’s diagnosis of GBS. A total of 11 cases reported overlaps of GBS variants. Four cases reported an overlap of the AIDP and AMAN variants, four cases reported an overlap of the AIDP and the AMSAN variant, one case reported the overlap of MFS and AMSAN, one case reported the overlap of AIDP and MFS, and one case reported the overlap of AMSAN and FD. The AIDP variant was the most prominent subtype and comprised 50.4% (*n* = 58) of the studies. The second most prominent variants were the axonal variants, including AMAN and AMSAN. The AMSAN variant made up 16.5% (*n* = 19) of the classification, while the AMAN variant made up 9.6% (*n* = 11). The MFS variant affected a total of seven patients (6.1%). The least common variants were the FD and the PCB variants of GBS, comprising 5.2% (*n* = 6) and 2.6% (*n* = 3), respectively. The distribution of GBS subtypes is shown in Table 3. 

## 4. Discussion

In our study, we observed that the majority of the patients with concomitant COVID-19 and GBS were male and over 40 years old. This observation is similar to the factors associated with COVID-19. A meta-analysis revealed a 3.5% increase in the disease severity per age year when measuring the relative risk estimate associated with age-related risk factors of COVID-19 severity [9]. Furthermore, males were suggested to have an increased susceptibility to the binding of the SARS-CoV-2 spike (S) glycoprotein and the angiotensin-converting enzyme 2 (ACE2) receptors on host cells, causing a downregulation in ACE2 [10]. Downregulation of ACE2 can be detrimental to patients that may already be deficient in ACE2. The male mortality rate for COVID-19 is influenced by the location of ACE2 on the X chromosome and how it influences an increased binding affinity between SARS-CoV-2 S glycoprotein and ACE2, according to Gadi et al. [10]. Men are more likely to contract COVID-19 and have a relatively higher risk of acquiring severe COVID-19 symptoms once they are already hospitalized and more often require ICU admission [11]. An epidemiological study in Finland reported that 57% (*n* = 559) of patients with GBS were male [12]. In our study, more male patients were reported to have GBS and COVID-19, supporting the earlier reports. 

According to our study, approximately half of the patients were in the hospital for 20 or more days (46%) and required admission to the ICU (44.9%), and 38% required MV indicating that these patients were severely affected by COVID-19 and GBS. Patients with GBS are historically prone to respiratory failure due to progressive respiratory muscle weakness, which causes a restrictive respiratory pattern [13]. Teitelbaum and Borel reported that about one-third of patients with GBS require MV and ICU admission due to respiratory failure [13]. 

Coronaviruses are thought to cause GBS directly through the neuroinvasive capacity of SARS-CoV-2 or as an autoimmune response triggered by a CRS mediated by the inflammatory response associated with COVID-19. Reports show that GBS associated with COVID-19 differs from the typical “post-infectious” pattern of GBS and presents more commonly as an “acute para-infection” [6]. The apparent difference between these infection patterns is that most infectious agents typically associated with GBS, such as varicella-zoster virus and cytomegalovirus, cause direct damage to the nerve roots due to the presence of the virus in the CSF, which appears unlikely in COVID-19 infections [6]. 

Cerebrospinal fluid analysis revealed that protein levels in CSF were elevated in 92% of the patients (*n* = 92). Our study results support previous reports of COVID-19 CSF analysis, showing that CSF protein was high in almost three-quarters of patients with severe and non-severe symptoms [14]. This study also found that CSF levels were elevated in all patients that presented with both CSF and COVID-19 [14]. These findings suggest that elevated CSF protein levels in COVID-19 patients can be a marker for CNS involvement. The classic immunologic alteration of CSF, albuminocytological dissociation, was described by Guillain, Barré, and Strohl in 1916 [15]. Cerebrospinal fluid albuminocytological dissociation was observed in 83.5% (*n* = 71) of cases. The CSF glucose values were abnormal in 52.9% (*n* = 18) of the 34 cases. High levels of CSF glucose were found to be significantly associated with severe impairments from GBS [16]. This factor is especially true in patients with diabetes mellitus or hyperglycemia, where the blood–brain barrier disruption is increased [16,17]. The overall findings from our data and previous literature attribute CSF findings such as increased protein, albuminocytological dissociation, and increased glucose as an indication of blood–CSF barrier disruption [16,18,19]. 

Our analysis observed the absence of SARS-CoV-2 in CSF when evaluating the results of cases that reported CSF PCR testing. Only four out of 55 cases (~7%) detected the presence of COVID-19 in the CSF. Similar results for COVID-19 in CSF were reported in studies on cases with a COVID-19 diagnosis (without GBS). Lewis et al. reported that the CSF SARS-CoV-2 PCR resulted in positive for 17/303 (6%) patients [20]. Furthermore, Lersy et al. found that four patients (7%) had a positive SARS-CoV-2 RT-PCR result in CSF [21]. The presence of anti-ganglioside antibodies in serum analysis was evaluated because of the typical ganglioside mimicry traditionally associated with GBS. Anti-ganglioside antibody testing revealed that only 18.8% of patients tested positive for anti-ganglioside antibodies typically associated with GBS. Similar results by Hasan et al. showed that only one out of 26 patients tested positive for anti-ganglioside antibodies [22]. This trend differs from the classical GBS presentation associated with molecular mimicry and suggests that the CRS may significantly impact the onset of COVID-19-related GBS.

Mistaken attacks on myelin sheaths or axons (the nerve conduits for sending and receiving neural signals) cause signature symptoms of GBS, such as rapidly progressive ascending symmetrical weakness, paresthesia, and sensory disturbance [7,23,24]. Symptoms of GBS are highly variable with respect to the antecedent. The variability of GBS symptoms is credited to multiple factors, including the extent of sensory symptoms and weakness, the presence, distribution, and scope of cranial nerve deficits, ataxia, pain, and autonomic dysfunction [7]. 

Through neuronal retrograde dissemination, CNS infection can be induced via the cranial nerves by the infection of the epithelial cells in the oral mucosa, where levels of ACE2 are highly expressed and very susceptible to binding with SARS-CoV-2 [25]. The olfactory nerve is considered a shortcut for many viruses to gain access to the brain through the olfactory bulb, after that spreading to specific brain areas, including the brainstem and the thalamus [25,26]. Deficits to olfactory nerve function were reported in several cases. In our study, 17.7% of patients presented with ageusia (*n* = 26), and 15.6% reported anosmia as a symptom (*n* = 23). SARS-CoV-2 may also enter the CNS via retrograde axonal transport through other peripheral nerves, including the trigeminal nerve, which possesses nociceptive cells in the nasal cavity [27]. The virus may also gain access through the sensory fibers of the glossopharyngeal nerve (cranial nerve nine) and the vagus nerve (cranial nerve ten) [27,28,29]. SARS-CoV-2 and its effects on cranial nerves (such as the glossopharyngeal and vagus nerve) may contribute to dysphagia. Dysphagia was present in 20.4% of cases (*n* = 30). Cranial nerve deficits may also contribute to the symptoms associated with respiratory distress and failure due to disruption of the innervations of the respiratory tract and lungs. Several cases also reported signs of pathology related to the facial nerve (the seventh cranial nerve). Facial nerve involvement can manifest as facial weakness, plegia, or paresthesia, resulting in 42.2% (*n* = 62) of cases. 

Aphasia, a neurological language disorder caused by brain injury, was present in 12.9% (*n* = 19) of cases. The etiology of aphasia is most often attributed to stroke, but may also be caused by traumatic brain injury, dementia, and brain tumors [30]. In patients with COVID-19, aphasia is likely caused by encephalopathy and ischemic stroke [30,31]. Encephalopathy is a significant concern and risk in COVID-19 due to hypoxia, metabolic changes, and the CRS augmented by the SARS-CoV-2 virus [31].

It was found that 46.3% of cases reported ataxia as a symptom, and 84.4% of patients reported hyporeflexia or areflexia. The loss of deep tendon reflexes is one of the characteristic symptoms of GBS and was expected to have a high frequency among cases. The patients that were diagnosed with MFS had a higher prevalence of areflexia and ataxia, where all cases present in the study reported both as a symptom [26,32,33,34,35,36,37]. The high frequency of areflexia and ataxia was not a surprise, as the usual triad for MFS consists of acute onset of external ophthalmoplegia, ataxia, and loss of tendon reflexes [26]. 

Aside from hyporeflexia, areflexia, and limb impairment, our study patients’ most common neurologic manifestation was impaired somatic sensation. Somatic sensation impairment was reported in about three-quarters of cases (72.8%). Somatic sensation impairment is often accompanied by myalgia and lumbar pain and was reported in 23.8% of patients and 15% of patients, respectively. For electrophysiological tests investigating either CMAP or SNAP, any patient with impaired action potentials was considered to have impaired somatic sensation. Analysis revealed that 86.9% of patients had abnormal values in SNAP testing, while 99% had impaired CMAP.

A total of 24 patients were diagnosed with the AMSAN variant of GBS (this number includes five cases with an AMSAN overlap with AIDP or MFS). Twenty-two patients in this group had results for electrophysiological testing, and all 22 cases had impaired action potentials for both the SNAP and the CMAP tests, indicating the involvement of both the motor and sensory nerves. There was one case that reported an AMSAN and MFS overlap syndrome. A total of 15 patients were diagnosed with the AMAN variant of GBS (this number includes four cases with an AMAN overlap with AIDP). Nerve conduction studies showed that out of the 11 cases with electrophysiological testing results, ten patients had abnormal CMAP results, while only five showed impaired SNAP. It is important to note that two cases that reported abnormal SNAP were with AMAN and AIDP overlap syndrome. The most common variant of GBS among patients was AIDP, comprising about half the total cases (50.4%). A total of 10 cases presented an overlap of AIDP and another GBS variant. Four patients were diagnosed with an overlap of AIDP and AMAN; four were found to have an overlap between AIDP and AMSAN. One had an overlap of AIDP and MFS, and one patient had an overlap of AIDP and FD. The least common variants reported among cases were MFS, which comprised about 6% of cases; FD, which comprised about 5% of cases; and PCB, which comprised <3% of cases. 

Limb strength and sensation were measured in all the patients. It was found that 93.2% of patients reported abnormal lower limb function, while 22 cases presented complete lower limb plegia. It was found that 85.7% of patients reported abnormal upper limb function, while 20 cases presented complete upper limb plegia. The difference between strength and sensation in the upper and lower limbs can be explained by the fact that GBS affects patients in a progressive, ascending manner, where lower limbs are first affected by weakness or paresthesia and then the trunk and upper limbs. The differences in limb impairment also explain why more patients presented with lower limb plegia. Neurological manifestations that were not frequently reported included neck flexion weakness (present in ~8% of cases), abnormal plantar response (present in ~8% of cases), urinary difficulties (present in ~11% of cases), and fecal incontinence (present in ~4% of cases). The symptoms of GBS are often specialized to the subtype of GBS present; the most prevalent subtypes are AMSAN, AIDP, AMAN, and MFS. Other less common subtypes include paraparetic GBS, FD, PCB GBS, Bickerstaff brainstem encephalitis, polyneuritis cranialis, and acute autonomic neuropathy. Electrophysiologic studies characterize the specific subtype of GBS, which are crucial in determining the effective treatment per patient. 

## 5. Conclusions

While numerous peer-reviewed case reports and cases series of concomitant GBS with COVID-19 exist, there is a gap in knowledge on the correlation between the two, and the complete spectrum of clinical characteristics of COVID-19-related GBS remains unknown. Recent findings suggest that SARS-CoV-2 is a trigger for GBS, as it follows a similar para-infectious pattern as the other viral agents which contribute to the onset of GBS. Exploring the extent to which SARS-CoV-2 infection and GBS are related pathophysiologically is crucial in delivering the optimal treatment to patients suffering from this concomitant occurrence. Assessing the biomarkers, diagnostic parameters, and severity of injuries between cases of COVID-19-related GBS will provide better means to explore this relationship and patient management with improved outcomes. Thus, the present study provides oversight of the confounding factors and clinical outcomes of the patients with concomitant GBS and COVID-19.

## Figures and Tables

**Figure 1 medicina-58-01835-f001:**
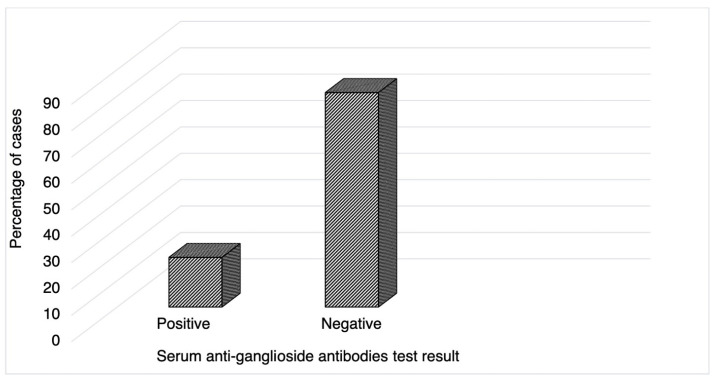
Serum anti-ganglioside antibodies test result.

**Figure 2 medicina-58-01835-f002:**
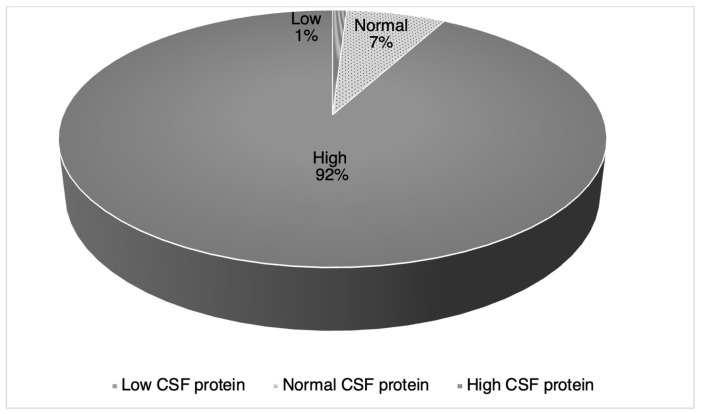
Cerebrospinal fluid protein values.

**Figure 3 medicina-58-01835-f003:**
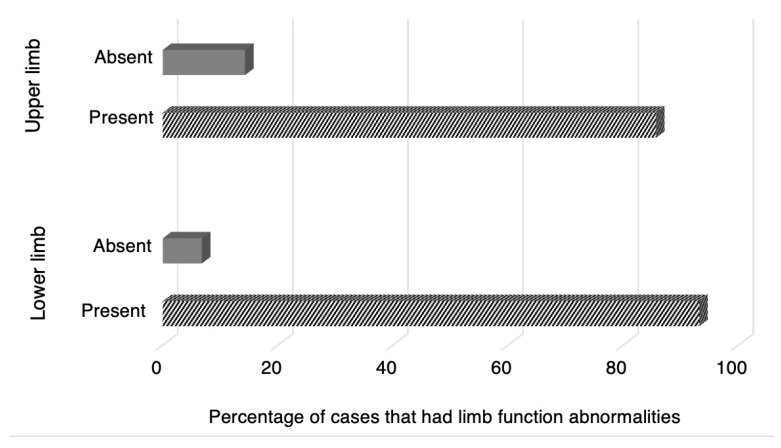
Percentage of cases with limb function abnormalities.

**Figure 4 medicina-58-01835-f004:**
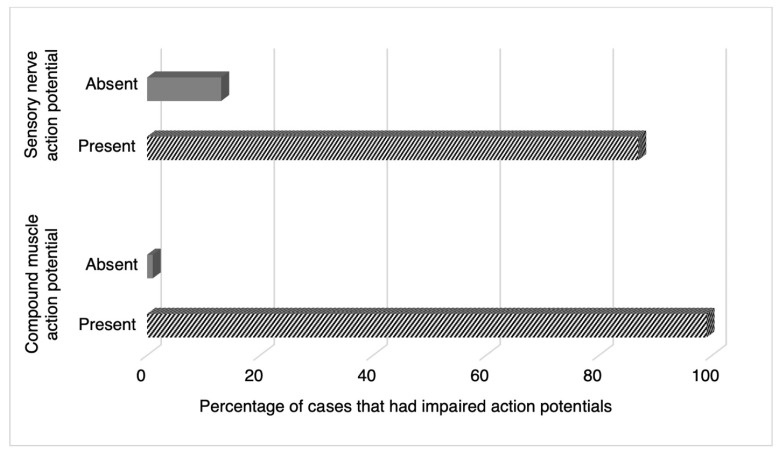
Sensory nerve and compound muscle action potential.

**Table 1 medicina-58-01835-t001:** Sociodemographic patient data and clinical outcome measures.

Category	(mean ± SD) [Range]	n	%
Age	52 ± 18 [3–94]		
Gender	Male	96	65.3
	Female	51	34.7
Age group	<40	33	22.4
	≥40	114	77.6
Hospital LOS (days)	23 ± 17 [1–62]		
Hospital LOS (days)	<20 days	30	54.5
	≥20	25	45.5
ICU admission	Yes	66	44.9
	No	81	55.1
Mechanical ventilation	Yes	56	38.1
	No	91	61.9
Number of days on mechanical ventilation	12 ± 7 [2–26]		
Mortality	Survived	116	78.9
	Deceased	16	10.9
	Hospitalized	15	10

SD: standard deviation; LOS: length of stay; ICU: intensive care unit.

**Table 2 medicina-58-01835-t002:** Neurophysiological findings.

Category	n	%
Abnormal plantar response	11	7.5
Aphasia	19	12.9
Ataxia	68	46.3
Dysphagia	30	20.4
Facial palsy/weakness/plegia	62	42.2
Fecal incontinence	6	4.1
Urinary difficulties	16	10.9
Hypogeusia/ ageusia	26	17.7
Hyporeflexia/ areflexia	124	84.4
Hyposmia/ anosmia	23	15.6
Impaired somatic sensation	107	72.8
Lumbar pain	22	15
Myalgia	35	23.8
Neck flexion weakness	11	7.5

**Table 3 medicina-58-01835-t003:** Distribution of Guillain–Barré syndrome subtypes.

GBS Subtype	n	%
Acute inflammatory demyelinating polyneuropathy (AIDP)	58	50.4
Acute motor axonal neuropathy (AMAN)	11	9.6
Acute motor-sensory axonal neuropathy (AMSAN)	19	16.5
Miller-Fisher syndrome (MFS)	7	6.1
Facial diplegia (FD)	6	5.2
Pharyngo-cervico-brachial (PCB)	3	2.6
AIDP and AMAN overlap	4	3.5
AIDP and AMSAN overlap	4	3.5
MFS and AMSAN overlap	1	0.9
AIDP and MFS overlap	1	0.9
AIDP and FD overlap	1	0.9

GBS: Guillain–Barré syndrome; AIDP: Acute inflammatory demyelinating polyneuropathy; AMAN: Acute motor axonal neuropathy; AMSAN: Acute motor-sensory axonal neuropathy; MFS: Miller-Fisher syndrome; FD: Facial diplegia; PCB: Pharyngo-cervico-brachial.

## Data Availability

Not applicable.

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
