# Peer review of "Concomitant Guillain–Barré Syndrome and COVID-19: A Meta-Analysis of Cases"

_medicina, 2022, doi:10.3390/medicina58121835_

Round 1

Reviewer 1 Report

row 172 - what is the relevance of the CSF glucose levels? if presented, must be commented

row 202 and 203, Table 3 and 4. - not clear enough. It may be redundant. Comments in the text are enough. 

row 296 - aphasia is a cortical sign, can not be contributed to the lesions of cranial nerves. Since you found aphasia in 12,9% of cases, please make a comment. 

row 337 - complete paralysis, meananing plegia?

Reviewer 2 Report

I read the Manuscript (Concomitant Guillain-Barré syndrome and COVID-19: A meta-analysis of cases) carefully. I still have some comments

1. In Abstract section Line 20 & 21:  ''The search was conducted through PubMed.'' why only Pubmed? Embase, Cochrane, Scopus and Web of Science.

2. In Abstract section Line 18: ''We have performed an analysis of 147 cases reported between April 2020-February 2022.'' belong to methods and not to background and objectives.

3. In Abstract section, Line 22 & 23: ''All case reports and series in the English language with accessed full text between 2020 and 2022 were included in the search'' please add the exact Date; Day and Month.

4. In abstract section: unter results pelase write the number of cases found after the search. 

5. In abstract section lines from 24-30: Please reorganise the text, demographic, clinical, laboratory, variants, Mortality. I mean realistic course of Data related to the practice. 

6. Please check this paper with the latest Syst. review and Metaanalysis in the topic: GUILLAIN BARRE SYNDROME FOLLOWING COVID-19 VACCINATION. A SYSTEMATIC REVIEW AND META-ANALYSIS OF CASE REPORTS AND CASE SERIES. - SHM Abstracts | Society of Hospital Medicine

7. In the methods section line 92-93. Table 1 show the  Sociodemographic patient data and clinical outcome measures' and not the reference values right?

8. the first paragraph in results section is not need as it is already written in the PRISMA

Round 2

Reviewer 2 Report

Thank you, the authors addressed all my comments.